# An Unsupervised Method for Industrial Image Anomaly Detection with Vision Transformer-Based Autoencoder

**DOI:** 10.3390/s24082440

**Published:** 2024-04-11

**Authors:** Qiying Yang, Rongzuo Guo

**Affiliations:** College of Computer Science, Sichuan Normal University, Chengdu 610101, China; gyz00001@163.com

**Keywords:** anomaly detection, Vision Transformer, memory network, attention mechanism

## Abstract

Existing industrial image anomaly detection techniques predominantly utilize codecs based on convolutional neural networks (CNNs). However, traditional convolutional autoencoders are limited to local features, struggling to assimilate global feature information. CNNs’ generalizability enables the reconstruction of certain anomalous regions. This is particularly evident when normal and abnormal regions, despite having similar pixel values, contain different semantic information, leading to ineffective anomaly detection. Furthermore, collecting abnormal image samples during actual industrial production poses challenges, often resulting in data imbalance. To mitigate these issues, this study proposes an unsupervised anomaly detection model employing the Vision Transformer (ViT) architecture, incorporating a Transformer structure to understand the global context between image blocks, thereby extracting a superior representation of feature information. It integrates a memory module to catalog normal sample features, both to counteract anomaly reconstruction issues and bolster feature representation, and additionally introduces a coordinate attention (CA) mechanism to intensify focus on image features at both spatial and channel dimensions, minimizing feature information loss and thereby enabling more precise anomaly identification and localization. Experiments conducted on two public datasets, MVTec AD and BeanTech AD, substantiate the method’s effectiveness, demonstrating an approximate 20% improvement in average AUROC% at the image level over traditional convolutional encoders.

## 1. Introduction

With the improvement of national living standards and the rapid development of the domestic manufacturing industry, the variety and quantity of industrial products have increased dramatically. This has elevated consumer expectations regarding product quality, particularly in terms of appearance. Currently, appearance quality is at the forefront of market competitiveness, necessitating meticulous management of product appearance in industrial production. Manufacturing processes face challenges from various defects, including porosity, fractures, and wear, which impact not only appearance but also performance and market competitiveness. Hence, implementing effective anomaly detection in manufacturing processes is crucial to ensuring production safety, maintaining product quality, and enhancing competitiveness. In practice, industrial anomaly detection demonstrates significant application value [1,2]. The goal of industrial image anomaly detection is to identify nonconforming products from images, safeguarding quality and enhancing economic returns. This task is challenging because anomalies often occupy only a small portion of the image, and industrial images are characterized by high data dimensionality, making feature extraction and anomaly detection and localization difficult. Thus, exploring effective anomaly detection methods to enhance efficiency remains a significant challenge.

Traditional image anomaly detection [3,4] is predicated on the manual definition and extraction of features, a process that is both cumbersome and time-consuming, particularly with extensive industrial image data. The rapid advancements in deep learning within artificial intelligence, notably in natural language processing and computer vision, have expanded its application to industrial anomaly detection [5]. Deep learning anomaly detection techniques are categorized based on the availability of data labels into fully supervised, unsupervised, and semi-supervised methods. Fully supervised learning methods [6,7], used in deep learning, depend on datasets meticulously labeled by experts for model training. Although effective, this approach poses a significant financial burden on many facilities due to the high labor costs involved. Confronted with the challenge of high data labeling costs, researchers have advocated for semi-supervised learning as a feasible solution. Semi-supervised learning employs a modest amount of labeled data to direct model training, enhancing the model’s generalization capability by incorporating substantial volumes of unlabeled data, making it particularly apt for scenarios with limited labeling resources or high labeling costs. For instance, Qiu et al. [8] developed a framework featuring an image alignment module and a defect detection module for identifying defects on metal surfaces. Wan et al. [9] introduced a Logit Inducing Loss (LIS) for training with imbalanced data distributions, alongside an Anomaly Capture Module to characterize anomalies, efficiently leveraging limited anomaly data. Despite their effectiveness in certain contexts, supervised learning models see limited use in industrial image anomaly detection. Collecting a substantial volume of anomaly samples for training supervised models is challenging in practice, with the labeling process being both time-consuming and expensive. While semi-supervised learning can mitigate labeling costs to some extent, it does not address the underlying issue. Conversely, unsupervised methods, which require only normal samples for training and eschew detailed labeling of abnormal samples, theoretically can identify all unknown defects, rendering them ideal for anomaly detection.

In unsupervised anomaly detection, deep codec structures based on convolutional neural networks (CNNs) are commonly employed, notably the Convolutional AutoEncoder (CAE) [10], which compresses a normal image and reconstructs it to resemble the original. However, the limited sensory field of traditional CNNs constrains CAE to learning merely local information, hindering the capture of global contextual image information, resulting in inferior image reconstruction quality. Furthermore, while the CAE’s widespread use in image reconstruction and anomaly detection owes to CNNs’ robust generalization capabilities, this generalizability can prove to be a drawback. Specifically, CAE may inadvertently reconstruct anomalous regions during image reconstruction, thereby diminishing anomaly detection accuracy. Recently, the application of the Transformer architecture [11] in computer vision [12] has garnered increasing research interest. This architecture, an encoder-decoder structure, leverages a self-attention mechanism to capture long-range dependencies in the input sequence, extracting global feature information. Such capability enables the Transformer to enhance processing efficiency and accuracy, maintaining global feature representation without dependence on traditional convolutional or Recurrent Neural Network (RNN) architectures. Mishra et al. [13] proposed a framework based on the Transformer for patch-level image reconstruction, utilizing Gaussian Mixture Density Networks to localize anomalous regions. Lee et al. [14] introduced AnoViT, an encoder-decoder model based on the Vision Transformer, asserting its superiority over CNN-based L2-CAE in anomaly detection. HaloAE [15] integrates the Transformer with HaloNet [16], facilitating image reconstruction and achieving competitive results on the MVTec AD dataset. Despite the Transformer’s exemplary performance and versatility in computer vision, the nuances of industrial image anomaly detection—namely, sensitivity to anomaly details and the scarcity of samples—necessitate focused optimization and enhancement of the architecture.

Given the challenges in practical industrial inspection applications—namely, the scarcity of anomaly samples, high labeling costs, and limitations of traditional CAE in extracting global features and controlling anomaly generalization—this study proposes an unsupervised industrial image anomaly detection method leveraging the Vision Transformer (ViT). The primary contributions of this study are as follows:Addressing the challenge of traditional CAE in learning global image features, the method incorporates a Transformer structure to understand the global context between image blocks, utilizing ViT’s encoder for high-level feature representation.To mitigate the issue of anomaly generalization that reduces detection accuracy, a memory module is designed to record the normal image features extracted by the encoder, suppressing anomaly generalization.A coordinate attention mechanism is introduced to concentrate on image features at both spatial and channel levels, enabling more precise anomaly localization and identification.L2 loss, block-based SSIM loss, and entropy loss functions are employed to define the relationship between original and reconstructed images and calculate the anomaly score, thereby enhancing detection accuracy.

## 2. Related Work

### 2.1. Unsupervised Anomaly Detection Method Based on Reconstruction

Within the realm of deep learning, unsupervised industrial anomaly detection methods are primarily divided into feature-based and reconstruction-based approaches. Feature-based approaches [17,18,19] commonly employ networks pre-trained on ImageNet [20] to map original images into a more distinguishable feature space. These methods are typically characterized by low training costs and outstanding performance. However, these approaches struggle to extract specific anomalous feature information from abstract feature vectors. Reconstruction-based approaches [21] are more prevalent, based on the assumption that models trained on normal samples excel in reconstructing normal patterns but falter with abnormal regions. The fundamental concept involves reconstructing the input normal image through encoding and decoding, training the neural network for this specific purpose. During the detection phase, the trained network identifies abnormalities by analyzing discrepancies between images pre- and post-reconstruction. A frequently employed metric involves calculating the L_2_ distance or SSIM value [22] between the images pre- and post-reconstruction, namely, the reconstruction error. Compared to feature-based approaches, reconstruction-based methods offer more intuitive visual comprehension by permitting the direct observation of differences between original and reconstructed images. Classic examples of reconstruction-based methods encompass auto-encoder (AE) [23], variational auto-encoder (VAE) [24] and generative adversarial network (GAN) [25].

Currently, most GAN-based and AE-based methods rely on convolutional neural networks (CNNs), which often perform suboptimally due to challenges in controlling the model’s generalization ability. When the generalization ability of AEs is excessively high, it may lead to a blending of normal and abnormal features, where an overgeneralized model reconstructs abnormalities too effectively, rendering them harder to distinguish. To counteract the effects of excessive generalization, recent strategies aim to constrain the expressive capabilities of the AE’s latent space, incorporating techniques such as memory storage [26], selection and weighting within the latent space [27], and and clustering of features therein [28]. However, weaker generalization capabilities lead to poorer reconstruction of edge regions. However, diminished generalization capabilities result in inferior reconstruction of edge regions. For instance, the VAE-based model FAVAE [29] tends to overlook image details during processing, which leads to issues of over-detection.

In this study, we employ a reconstruction-based approach for image feature extraction, utilizing Transformer-based encoders instead of CNN-based encoders. Furthermore, diverging from methods that utilize the entire image as input, this study segments the image into uniformly sized chunks for model input, and further extracts the image’s rich features by learning global information through patch-level images and self-attention mechanisms.

### 2.2. Vision Transformer

The Vision Transformer (ViT) [30] represents an innovative application of Transformer structures to image processing tasks, marking a significant breakthrough in the field of image recognition and detection. In this chapter, a ViT-based encoder is utilized to supplant the traditional convolutional encoder, beginning with an overview of the ViT model’s workflow. Figure 1 illustrates the ViT model structure, comprising the Embedding Layer, Transformer Encoder, and MLP Head.

Embedding Layer

As depicted in Figure 1a, ViT initially divides the image into multiple fixed-size patches, embedding positional information for each patch before inputting this data into the Transformer Encoder (Figure 1b), equipped with the self-attention module.

2.Transformer Encoder

The image patches, now transformed into a series of vectors by the embedding layer, are introduced into the encoder component of the Transformer. As illustrated in Figure 1b, the Transformer Encoder comprises a series of L identical layers, encoding the input image into a high-dimensional feature representation. Each layer features Layer Normalization (LN), a Multi-head Self-Attention Mechanism (MSA), and a Multi-Layer Perceptron (MLP).

3.MLP Head

The high-dimensional feature vectors produced by the Transformer Encoder require appropriate transformation for classification tasks. For downstream classification tasks, the model extracts features aligned with the category labels’ quantity, serving as the foundation for classification. Hence, the model’s output phase typically utilizes fully connected layers and activation functions to map the feature space to the label space for classification.

The self-attention mechanism [31] represents a crucial component within the Transformer architecture, dynamically adjusting the representation of each sequence element. This mechanism generates a weighted representation for each position, considering the interactions among elements within the input sequence. It is particularly adept at capturing long-range dependencies within a sequence. The computational process of self-attention encompasses three principal sets of vectors: query (***Q***), key (***K***) and value (***V***), as depicted in Equation (1):(1)$$\mathrm{Attention}(Q,K,V)=\mathrm{softmax}(\frac{Q.K^{T}}{\sqrt{d_{k}}})V$$

In Equation (1), ***Q*** is the query matrix; ***K*** is the key matrix; ***V*** is the value matrix; *d_k_* denotes the dimension of ***K***; $\sqrt{d_{k}}$ is the scaling factor used to scale the dot product in order to keep the gradient stable; and *T* denotes the transpose operation of the matrix. The softmax function is used to calculate the attention weight.

The multi-head attention mechanism in ViT represents an advanced refinement of the self-attention layer, enabling simultaneous information capture across multiple subspaces and thus enhancing the model’s representational capabilities. This mechanism segments the self-attention layer into multiple “heads,” with each performing independent self-attention operations using distinct weight matrices. Consequently, this enables the model to concentrate on various parts or aspects of the input sequence, thereby capturing richer and more varied information, as demonstrated in Equations (2) and (3):(2)$${\mathrm{head}}_{i}=\mathrm{Attention}(QW_{i}^{Q},KW_{i}^{K},VW_{i}^{V})$$
(3)$$\mathrm{MultiHead}(Q,K,V)=\mathrm{Concat}({\mathrm{head}}_{1},\cdots ,{\mathrm{head}}_{n})W^{o}$$
here, *n* is the total number of heads, $W_{i}^{Q}$, $W_{i}^{K}$, $W_{i}^{V}$ denote the linear transformation weight matrices for the *i*-th query, key, and value matrices (***Q***, ***K***, and ***V***), respectively, with $W^{o}$ being the output transformation matrix. The *h* function executes the computation for the *i*-th attention head, the *C* function aggregates the values across each attention channel, and the *M* function calculates the multi-head self-attention value.

## 3. Proposed Method

The method proposed in this study leverages the benefits of both reconstruction-based and patch-based approaches. The model comprises a ViT-based encoder, a memory module (M), a coordinate attention (CA) mechanism, and a decoder, as depicted in Figure 2. The specific workflow is as follows: 

**Input image processing:** Initially, the input image $X\in R^{H\times W\times C}$ is decomposed into patches $X_{p}\in R^{N\times (P^{2}\times C)}$ of size (*P*, *P*). These patches are then mapped into a D-dimensional space to form two-dimensional patch embeddings. Here, *H*, *W*, and *C* denote the height, width, and number of channels of the image, respectively, while *N* = *HW*/*P*^2^ represents the total number of patches.

**Position embedding and sequence encoding:** To preserve spatial location information among patches, position embedding $E_{pos}\in R^{(N+1)\times D}$ is added to each patch embedding *E*, and a [*cls*] token is introduced for global image representation.

**Encoder processing:** The encoder processes the encoded sequence *Z*_0_ and outputs a series of block embeddings, each detailing information about the corresponding image block. These block embeddings are rearranged to form a feature mapping map F that approximates the structure of the original image.

**Memory module:** The feature mapping map F is input into the memory module, generating a feature representation of the normal data pattern through matching and weighted summation with stored normal pattern features. This step allows the model to “remember” the normal data patterns.

**Coordinate Attention:** The feature maps F’ computed by the memory network are conveyed to the CA module for further processing, yielding enhanced feature representations.

**Decoding and image reconstruction:** Finally, the decoder reverses the high-dimensional feature representation from the coordinate attention module, rendering it into a reconstructed image. Comparing differences between the input image *X* and the reconstructed image $\hat{X}$ enables anomaly detection and localization.

### 3.1. ViT-Based Encoder

In the model, the encoder employs a processing flow akin to the Vision Transformer (ViT), transforming the input image patches into a series of high-dimensional feature representations for subsequent processing and analysis. 

The input image patches are initially mapped to the embedding space and augmented with positional information, as described in Equation (4):(4)$$Z_{0}=[X_{p}^{1}E;X_{p}^{2}E;\ldots ;X_{p}^{N}E]+E_{\mathrm{pos}}\;,\;E\in R^{(P^{2}\times C)\times D}\;,\;E_{\mathrm{pos}}\in R^{(N+1)\times D}$$

Subsequently, the patch embedding sequences, augmented with location information, are fed into the Multihead Self-Attention (MSA) module (Equation (5)) and the Multilayer Perceptron (MLP) module (Equation (6)), where Layer Normalization (LN) is applied to the feature sequences before they are passed to the MSA and the MLP, a step that aids in stabilizing the training process, with *l* (*l* ∈ [1,*L*]) denoting the number of layers:(5)$$Z_{l}^{\prime }=\mathrm{MSA}(\mathrm{LN}(Z_{l-1}))+Z_{l-1}\;,\;l=1\ldots L$$
(6)$$Z_{l}=\mathrm{MLP}(\mathrm{LN}(Z_{l}^{\prime }))+Z_{l}^{\prime }\;,\;l=1\ldots L$$

Finally, residual connections are introduced to the outputs of the MSA and MLP modules, meaning the module inputs are directly added to their outputs. This approach mitigates the issue of vanishing gradients during deep network training and enhances the network’s training stability. Through these steps, the encoder efficiently extracts and processes the features of the input image patches, generating a rich feature representation. Additionally, this process ensures the model’s ability to fully leverage the spatial information and intrinsic features of the image, thereby enhancing overall performance.

### 3.2. Memory Network

The memory module, a pivotal component, is designed to bolster the model’s capacity for memorizing normal data patterns, thereby aiding in more accurate recognition of deviations from these patterns. This functionality is realized by storing and processing the query mapping *z* extracted from the feature map F, output by Encoder, with its network structure depicted in Figure 3. In essence, the memory module is represented by a matrix $M\in R^{N\times C}$, where *M* is an addressable memory matrix, *N* denotes the number of feature information items in the memory, and *C* represents the dimensionality of each feature vector. This matrix functions as a long-term memory bank, storing valid feature information acquired during the training on normal data. Each element $m_{i}$ (where i = 1, 2, 3...N) in the memory *M* represents the i-th memory item, with each item being a C-dimensional feature vector representing a specific normal data pattern learned during training.

To access the memory item, the similarity between the input feature *z* and the memory item $m_{i}$ in the *M* is initially calculated:(7)$$d(z,m_{i})=\frac{zm_{i}^{T}}{\left\|z\right\|\left\|m_{i}\right\|}$$

In Equation (7), d(•,•) is defined as the cosine similarity [32]. Subsequently, a softmax operation is applied to the similarity scores between *z* and all memory items in ${\left\{m_{i}\right\}}_{i=1}^{N}$, yielding the weight coefficient *w* (*w*∈R^1×C^) for the input feature *z* with respect to the memory storage unit *M*:(8)$$w_{i}=\frac{\exp(d(z,m_{i}))}{\sum_{j=1}^{N}\exp(d(z,m_{j}))}$$

The weighting coefficients indicate the degree of match between the current feature and the memory items in *M*. Memory items corresponding to the input feature *z* are retrieved, weighted, and summed to acquire a new feature representation $\hat{z}$:(9)$$\hat{z}=wM=\sum_{i=1}^{N}w_{i}m_{i}$$

To mitigate the issue of certain anomalies being reconstructible during the model’s reconstruction process and to minimize interference from excessive irrelevant memory terms, a sparsification operation can be applied to the weights *w*:(10)$${\hat{w}}_{i}=\frac{\max(w_{i}-\lambda ,0).w_{i}}{\left|w_{i}-\lambda \right|+\varepsilon }$$
where, the shrinkage threshold *λ* is set to 1/N (N represents the number of memory items), and *ε* is a very small positive number to prevent division by zero in subsequent calculations and ensure numerical stability. The weights *w* are processed through a shrinkage function to diminish the values of less significant weights. The contraction function max(⋅,0) here refers to the ReLU activation function. The processed weights then undergo a normalization operation ${\hat{w}}_{i}=\frac{{\hat{w}}_{i}}{{\left\|{\hat{w}}_{i}\right\|}_{1}}$ to ensure their sum equals 1. This step aims to maintain weight distribution consistency and ensure the model output’s stability. Finally, the sparsified and normalized weights are utilized to calculate the weighted sum of all memory terms, generating the final feature representation $\hat{z}=\hat{w}M.$ This feature representation will thus be more focused on the most critical and useful information for the current input.

### 3.3. Coordinate Attention

Coordinate Attention (CA) [33] is a lightweight attention mechanism designed to enhance feature representation. It can process any intermediate feature tensor $X=[x_{1},x_{2},\cdots ,x_{c}]\in R^{C\times H\times W}$ within the network, transforming it into a tensor $Y=[y_{1},y_{2},\cdots ,y_{c}]\in R^{C\times H\times W}$ with identical size and dimensions, with its structure illustrated in Figure 4. 

Specifically, the input feature maps undergo a 1D global average pooling operation along the horizontal (H, 1) and vertical (1, W) directions, respectively, aggregating features in both spatial directions to form feature maps that capture the long-distance dependencies along each spatial direction. The resulting feature maps are then encoded to emphasize the weights of target regions within the original feature maps. This mechanism integrates coordinate information into channel attention, capturing long-range dependencies while preserving precise positional information, facilitating more accurate target location and recognition by the model. The Coordinate Attention (CA) module is positioned between the memory module and the encoder, with the feature map *F*′ processed by the memory module passed onto the CA module, enabling further feature enhancement. The enhanced feature representations are then directed to the decoder for image reconstruction, whereby the decoder, leveraging the rich and precise feature information from the CA module, reconstructs the image with greater accuracy, particularly in the presence of subtle abnormalities. Furthermore, the CA module is flexible, with a minimal parameter count, allowing for easy integration into network modules to achieve notable performance improvements without substantial computational resources.

### 3.4. Decoder

The decoder is tasked with inversely mapping the high-dimensional feature representations processed by the encoder or the memory module back into the image space, thereby generating the reconstructed image. As depicted in Figure 2, the decoder incrementally increases the size of the feature map using six transposed convolutional layers, each consisting of two sublayers: a normalization layer and a ReLU layer. The normalization layer normalizes small batches of data, accelerating the training process and enhancing model stability, while the ReLU function introduces non-linear processing capabilities, enabling the model to capture complex input-output relationships. Applying ReLU after each normalization layer allows the decoder to more effectively learn and reconstruct image details. The final layer employs tanh as the last non-linear activation function. Through these operations, the decoder inversely maps the high-dimensional feature vectors back into the image space, generating a reconstructed image X^∈ℝH×W×C akin to the original input image. Differences between the reconstructed and original input images are utilized to assess the model’s performance, with anomalous regions often efficiently identified through significant differences in the reconstructed image.

### 3.5. Loss Functions

In this study, normal images serve as training samples, with both normal and abnormal images being input into the model during the testing phase to attempt data reconstruction. Given the model’s learning and memorization of normal image patterns, the use of abnormal images as inputs results in poor reconstruction of abnormal regions, leading to significant reconstruction errors. To ensure the model’s predictions closely align with actual values, multiple loss functions are employed in training. The loss function utilized in this study comprises two components: image reconstruction loss and entropy loss associated with the query weights in the memory module.

1.Reconstruction loss function

Reconstruction loss comprises two components: per-pixel mean squared error loss $L_{2}$ (Equation (11)) and block-based Structural Similarity (SSIM) loss [14] (Equation (12)). Per-pixel mean squared error loss quantifies pixel-level differences between the reconstructed and original images, while block-based SSIM loss evaluates similarity in structure and texture, ensuring visual consistency between the reconstructed and original images.
(11)$$L_{2}(X,\hat{X})=\frac{1}{WH}{\|X-\hat{X}\|}_{2}^{2}$$
(12)$$L_{SSIM}(X,\hat{X})=\frac{1}{WH}\sum_{i=1}^{H}\sum_{j=1}^{W}1-SSIM{\left(X,\hat{X}\right)}_{\left(i,j\right)}$$
where *X* and $\hat{X}$ denote the original and reconstructed images, respectively, with *H* and *W* representing their height and width, and *W* × *H* indicating the total number of pixels, and $SSIM{\left(X,\hat{X}\right)}_{\left(i,j\right)}$ represents the SSIM value for two corresponding blocks in *X* and $\hat{X}$ centered at coordinates (i, j). Finally, the reconstruction loss is defined as follows:(13)$$L_{recon}(X,\hat{X})=L_{2}(X,\hat{X})+\lambda L_{SSIM}(X,\hat{X})$$

Here, *λ* is a hyperparameter utilized to balance the contributions of the two types of loss: mean squared error loss and SSIM loss.

2.Entropy loss function

In the memory module, the entropy loss on the matching probability *w_i_* of each memory item assists in evaluating the model’s utilization of various memory items, thus optimizing memory utilization efficiency and the model’s generalization ability, defined as follows:(14)$$L_{ent}=\sum_{i=1}^{N}{\hat{w}}_{i}\log({\hat{w}}_{i})$$
where ${\hat{w}}_{i}$ represents the matching probability of the *i*-th memory item in the memory module. Minimizing the entropy loss encourages the model to adjust the matching probability of certain memory items close to 1 and others near 0, indicating increased model “confidence” in selected memory items and reduced decision-making uncertainty. 

To balance the reconstruction loss and entropy loss, a comprehensive loss function is constructed for model training:(15)$$Loss=\lambda_{recon}L_{recon}+\lambda_{ent}L_{ent}$$
where *L_recon_* represents the reconstruction loss, assessing the similarity between the reconstructed and original images; *L_ent_* represents the entropy loss, quantifying the model’s certainty regarding the probability of matching memory items; and *λ_recon_* and *λ_ent_* are the weight parameters regulating the impact of the respective losses on the total loss.

## 4. Experiments and Results

### 4.1. Datasets

This study employs the MVTec AD dataset [22] and the BeanTech AD dataset [13] for anomaly detection. These datasets are extensively utilized to evaluate anomaly detection algorithms, offering diverse testing scenarios and data for this research.

MVTec AD

Released by MVTec in 2019 at the CVPR conference, the MVTec AD dataset is tailored for unsupervised image anomaly detection tasks. Simulating real-world industrial scenarios, its comprehensiveness and practicality have led to widespread use. The dataset comprises a broad collection of object categories, a rich variety of anomaly types, and pixel-level labels for anomalous images. Encompassing 5354 high-resolution color images (3629 normal and 1725 anomalous), resolutions are either 700 × 700 or 1024 × 1024 pixels. Spanning 15 categories (5 texture and 10 object), the dataset covers over 70 types of artificially induced defects, such as scratches, stains, and deformations. Each category includes about 60–400 normal training samples, along with a mix of normal and anomalous test images. Table 1 summarizes the dataset’s key statistical data, where N denotes the number of normal samples, and P the number of anomalous samples.

BeanTech AD

The BeanTech AD dataset, designed for unsupervised anomaly detection in industrial settings, features a structure similar to the MVTec AD dataset. It comprises 2830 RGB images across three different industrial products. Specifically, product 1 images are 1600 × 1600 pixels, product 2 images 600 × 600 pixels, and product 3 images 800 × 600 pixels. The training set includes 400 images for product 1, 1000 for product 2, and 399 for product 3. All images feature pixel-level labels indicating the location and size of anomalous regions, as illustrated in Figure 5.

### 4.2. Parameter Settings

This experiment comprises two phases: training and testing, with both phases initiated from scratch. Initially, the size of the input image is uniformly adjusted to 384 × 384, the segmented image block size is set to 16 × 16, the embedding dimension is set to 768 (representing the vector size mapping the image block to high-dimensional space), and the multi-head self-attention mechanism is configured with 8 heads. The model undergoes 200 training epochs, the batch size is set to 8, the learning rate is established at 0.0002, hyperparameters *λ_recon_* and *λ_ent_* in the loss function are set to 1.0 and 0.001, respectively, with remaining hyperparameters using the network’s default configuration. Additionally, the method employs the Adam optimizer for training and grid search for hyperparameter optimization. Gaussian smoothing serves as a post-processing technique for image anomaly detection, generating a final score map by distributing each pixel’s score in a Gaussian distribution, a technique applied in various studies [34,35,36]. Furthermore, this experiment is conducted using the PyTorch framework, with the specific experimental environment detailed in Table 2: 

### 4.3. Evaluation Metrics

This study employs two metrics to comprehensively assess the model’s performance. Firstly, the Area Under the Receiver Operating Characteristic (AUROC), at both image and pixel levels, serves as a commonly used primary standard for model assessment. AUROC, derived from False Positive Rate (FPR) and True Positive Rate (TPR) values, is a threshold-independent performance measure unaffected by the positive to negative sample ratio, making it especially suitable for datasets with imbalanced samples.

Image-level AUROC evaluates the model’s overall efficiency in anomaly detection, while pixel-level AUROC assesses the model’s precision in localizing anomalous regions. However, given that many anomalies occupy only a small portion of an image’s pixels, pixel-level AUROC may not fully capture the true performance of anomaly localization. Occasionally, despite a high number of false positives, AUROC can remain high if the FPR is low. Therefore, when the FPR is below 0.3, this study introduces the Per-Region Overlap (PRO) [17] as a supplementary metric for anomaly localization assessment. The calculation Equation for this metric is as follows:(16)$$\mathrm{PRO}=\frac{1}{N}\sum_{n=1}^{N}\frac{P\cap G_{n}}{G_{n}}$$

In the Equation, *N* denotes localized defect results, with actual true values divided into *N* regions based on connectivity, *P* represents predicted values, *G_n_* represents true values, and *P* ∩ *G_n_* denotes the intersection between predicted and true values in each region. Higher values of AUROC and PRO indicate superior model performance. AUROC measures the model’s overall efficiency in anomaly detection, whereas PRO more closely examines the model’s precision in localizing anomalous regions.

### 4.4. Comparative Experiment

#### 4.4.1. Results

To evaluate the performance of the aforementioned methods, a comparative analysis was conducted using existing and recently proposed techniques on the MVTecAD and BTAD datasets.

On the MVTecAD dataset, the proposed method was compared with various unsupervised learning algorithms, including AE_SSIM [37], AnoGAN [38], L2-CAE [14], VAE [39], MKD [40], CAVGA [41], SCADN [42], AnoViT [14], VT-ADL [13]. Among these, AE_SSIM [37], L2-CAE [14], and VAE [39] utilize a convolutional autoencoder architecture, while AnoViT [14] and VT-ADL [13] employ a Transformer-based architecture. Experimental results are detailed in Table 3 and Table 4, with Table 3 displaying each model’s performance on the image-level AUROC score, reflecting the models’ anomaly detection capabilities with an average score of 91.2%. Conversely, Table 4 details each model’s performance on anomaly localization, evaluated by pixel-level AUROC and PRO scores, with average scores of 93.0% and 92.3%, respectively, across the dataset.

On the BTAD dataset, our approach was compared against popular techniques like P-SVDD [43], SSPCAB [44], DRAEM [45], and VT-ADL [13]. Table 5 details the experimental results, covering the performance evaluation for anomaly detection and localization. The results indicate that the average image-level and pixel-level AUROC scores were 92.2% and 91.9%, respectively, with the average PRO score at 91.0%.

#### 4.4.2. Analysis

(1)MVTec AD

Anomaly detection results in Table 3 reveal that our method achieves the highest scores in 9 categories, notably hazelnut and screw, with a performance of 100%. Compared to traditional convolutional autoencoder-based methods (AE_SSIM, L2-CAE, VAE), our method’s overall average performance improved by 14% to 28%. Furthermore, anomaly localization scores in Table 4 indicate significant performance enhancement across all 15 categories, with average pixel-level AUROC values improving by 2% to 18.7% and average PRO values by 0.9% to 48%, affirming our method’s exceptional performance in both anomaly detection and localization.

In particular, the codec utilizing the ViT architecture significantly surpasses traditional autoencoder models in anomaly detection, owing to the ViT’s capability to retain spatial information of embedded patches and comprehend the image’s global context, enabling more effective differentiation between normal and anomalous conditions. Compared to the AnoViT method, also based on Transformer architecture, our method registers significant improvements in all categories. For anomaly localization, our method outperforms VT_ADL, which also utilizes ViT as a backbone, in almost all categories. This success is primarily attributed to the inclusion of the memory module, significantly enhancing the model’s capability to infer anomalous regions. However, detection results in challenging categories, like transistors, are not satisfactory, which may be due to the model’s difficulty in accurately identifying and localizing subtle anomalies, such as missing or misoriented transistors.

Additionally, Figure 6 displays visualization results on the MVTecAD dataset. From the figure, the method identifies anomalous regions closely resembling the ground truth mask and locates anomalies more effectively than L2_CAE and AnoViT, detecting anomalous regions across a range of industrial images, irrespective of size and shape.

(2)BTAD

Based on average metrics in Table 5, our method leads in overall dataset performance, with average image-level and pixel-level AUROC scores of 92.2% and 91.9%, respectively, and an average PRO score of 91.0%. Compared to other methods, our results show improvements of 3.6% to 10.5%, 0.2% to 10.1%, and 2% to 37% across each metric, respectively. Particularly, our method excels in anomaly detection and localization for product 01, achieving the highest scores across all evaluation metrics. Compared to VT_ADL, which also employs ViT as a feature extraction framework, our method outperforms in nearly all classification tasks, further affirming its effectiveness. Additionally, Figure 7 presents visualization results on the BTAD dataset, intuitively demonstrating our method’s capability to clearly identify and localize anomalous regions.

### 4.5. Ablation Experiment and Analysis

This subsection presents an ablation study conducted on the MVTecAD dataset to investigate the effectiveness of different workflow modules, with results displayed in Table 6. Table 6 details the average performance across 15 categories in terms of image-level AUROC, pixel-level AUROC, and PRO metrics. Specifically, a model utilizing only the ViT encoder and decoder serves as the base network (Base), with M representing the incorporated memory module and CA the coordinate attention mechanism.

Table 6 shows that the introduction of the memory module (M) and coordinate attention mechanism (CA) significantly enhances anomaly detection and localization. Adding only the M module to the base network increases image-level AUROC (I-AUROC), pixel-level AUROC (P-AUROC), and PRO metrics by 3.5%, 8%, and 20.9%, respectively. Conversely, adding only the CA module results in improvements of 2.3%, 2.6%, and 6.1% in these metrics, respectively. Introducing both M and CA modules improves I-AUROC to 91.2%, P-AUROC to 93.0%, and PRO to 92.3%. Overall, the M module significantly impacts pixel-level AUROC and PRO metrics, whereas introducing the CA module at various points yields different outcomes, notably when placed between the memory module and encoder, slightly enhancing overall anomaly detection and localization. Figure 8 offers a visualization comparison of the ablation study results on the MVTecAD dataset. Image comparison reveals that incorporating the M and CA modules significantly enhances model performance in anomaly detection, markedly increasing agreement between identified anomalous regions and ground truth (GT). This visualization intuitively demonstrates the effectiveness of both M and CA modules in enhancing model accuracy.

## 5. Conclusions

This study proposes an unsupervised industrial image anomaly detection method based on the Vision Transformer (ViT) to address the scarcity of anomalous image samples, high tagging costs in industry, limitations of traditional convolutional encoders in extracting global features, and anomalous generalization issues, utilizing a ViT-based encoder for feature extraction to effectively capture global contextual information, thereby enhancing detailed and holistic image understanding. The memory module stores feature information, enabling effective inference of abnormal regions and suppression of anomalous reconstruction. The introduced Coordinate Attention (CA) mechanism further enhances the model’s ability to capture image features, particularly at spatial and channel levels. By precisely focusing on important features and regions, the CA mechanism minimizes feature information loss and enhances anomaly recognition accuracy. The effectiveness of the proposed method has been fully validated through ablation studies on two public datasets and comparisons with contemporary mainstream models. Experimental results demonstrate the method’s strong detection performance and generalizability. However, the substantial content required for storage in the memory unit leads to increased computational demands. Future work will focus on reducing computational demands while further improving anomaly detection performance. Additionally, improving the clarity of stored potential features is essential, given the limitations of current memory module storage methods. Future efforts will aim to identify more efficient storage structures to enhance the memory module’s operational efficiency through improved query efficiency.

## Figures and Tables

**Figure 1 sensors-24-02440-f001:**
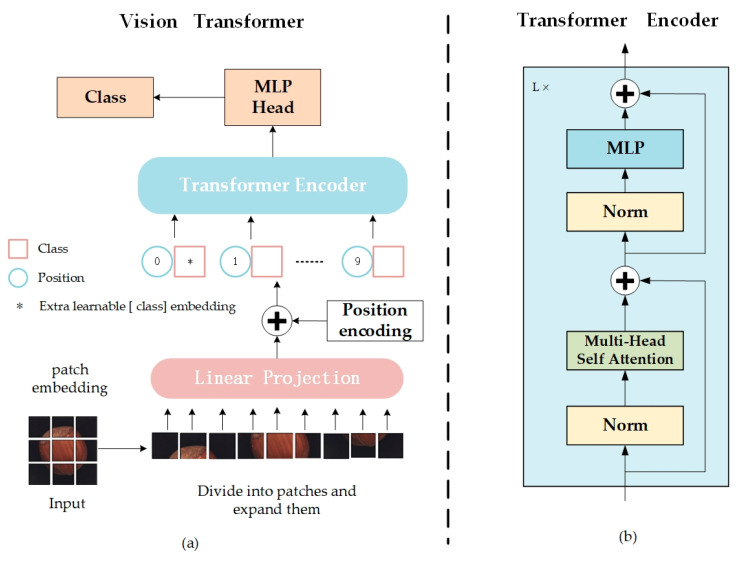
Architecture of original ViT. (**a**) Flowchart of vision transformer; (**b**) Structure of transformer encoder.

**Figure 2 sensors-24-02440-f002:**
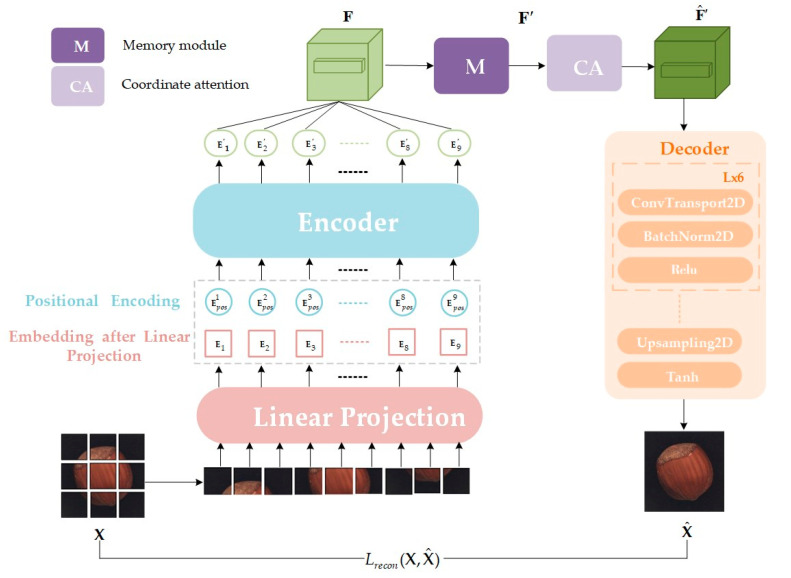
The flowchart of proposed method based on vision transformer and autoencoder structure. The model mainly consists of a ViT-based encoder, a memory module (M), a coordinate attention (CA), and a decoder. X and $\hat{X}$ represent the input image and the reconstructed output image, respectively.

**Figure 3 sensors-24-02440-f003:**
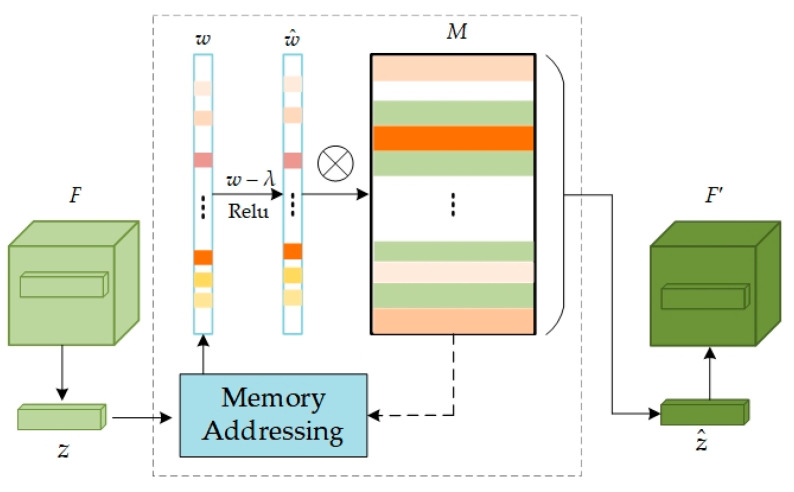
The network structure of memory modules M. *z* represents the input features extracted from the feature map and $\hat{z}$ represents the enhanced features computed by the memory module.

**Figure 4 sensors-24-02440-f004:**
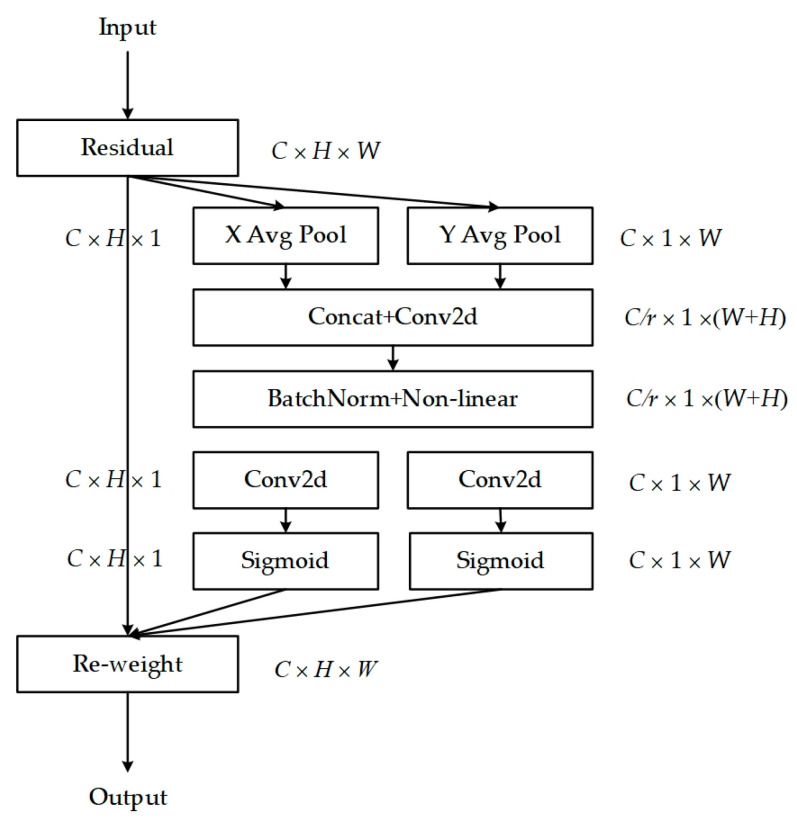
The flowchart of coordinate attention.

**Figure 5 sensors-24-02440-f005:**
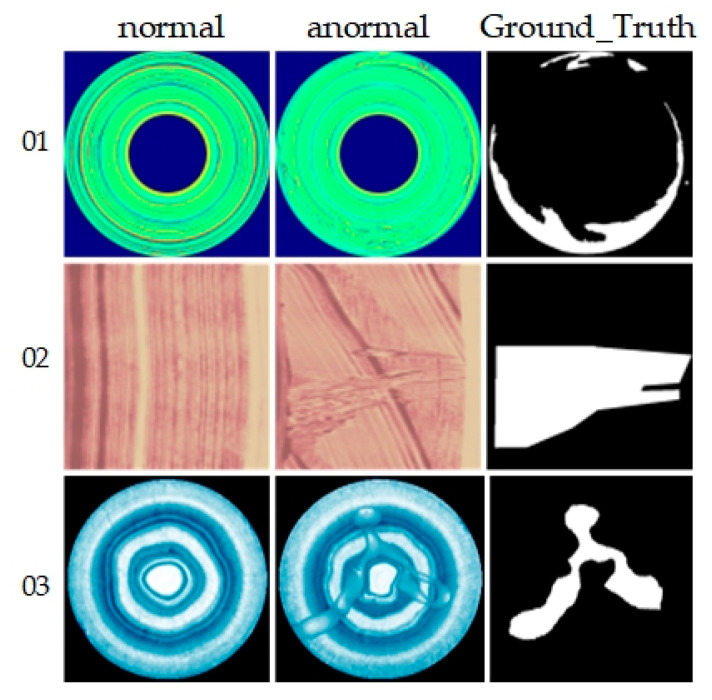
Examples of BTAD. From top to bottom are products 01, 02 and 03, representing normal images, abnormal images and ground truth from left to right.

**Figure 6 sensors-24-02440-f006:**
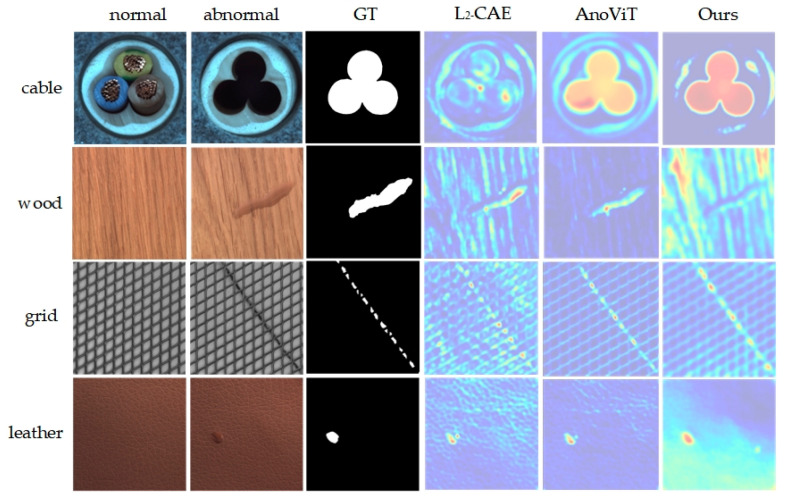
Visualization of MVTecAD dataset detection results. From top to bottom are the 4 categories of cable, wood, greed and leather, and from left to right are the input normal image, anomaly image, ground truth image and the results of L_2__CAE, AnoViT and the method in this paper.

**Figure 7 sensors-24-02440-f007:**
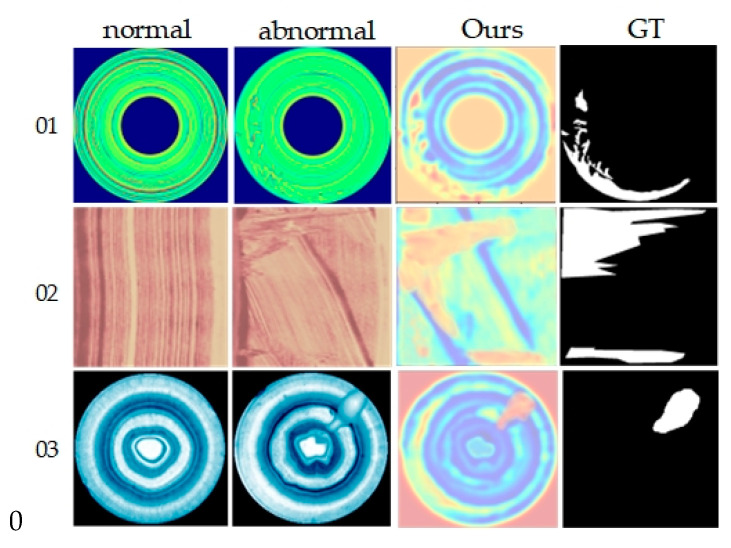
Visualization of BTAD dataset detection results. From top to bottom are products 01, 02 and 03, and from left to right are the input normal image, anomaly image, ground truth image and the results of the proposed method in this paper.

**Figure 8 sensors-24-02440-f008:**
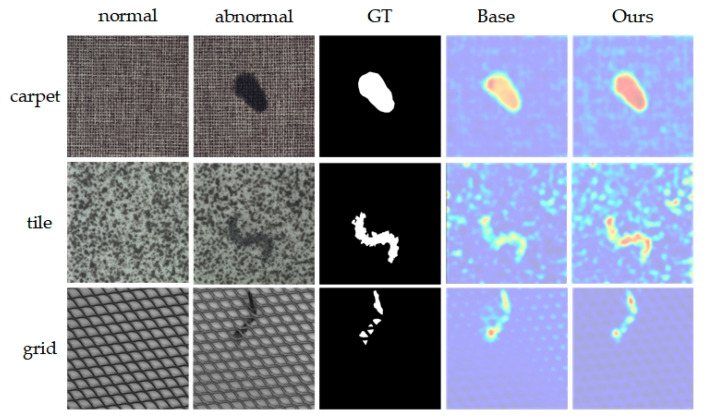
Visualization of ablation experiment results on MVTecAD. From top to bottom are the 3 categories of carpet, tile and grid, and from left to right are the input normal image, anomaly image, ground truth image and the results of Base and the proposed method in this paper.

**Table 1 sensors-24-02440-t001:** Detailed information of the MVTecAD.

	Category	Train	Test		Resolution Ratio
		N	N	P	
Texture	Carpet	280	28	89	1024
	Grid	264	21	57	1024
	Leather	245	32	92	1024
	Tile	230	33	84	840
	Wood	247	19	60	1024
Object	Bottle	209	20	63	900
	Cable	224	58	92	1024
	Capsule	219	23	109	1000
	Hazelnut	391	40	70	1024
	Metalnut	220	22	93	700
	Pill	267	26	141	800
	Screw	320	41	119	1024
	Toothbrush	60	12	30	1024
	Transistor	213	60	40	1024
	Zipper	240	32	119	1024
	Total	3629	467	1258	

**Table 2 sensors-24-02440-t002:** Experimental Environment Configuration.

Configuration Name	Environmental Parameters
Operating system	Windows 11
CPU	Intel(R) Xeon(R) Platinum 8358P
Memory	64 GB
GPU	NVIDIA GTX A5000 (24 G)
Programming	Python 3.8
CUDA	CUDA 11.0
Framework	Pytorch 1.7.1

**Table 3 sensors-24-02440-t003:** Anomaly detection results on the MVTec AD dataset (image level AUROC (%)).

Category	AE_SSIM	AnoGAN	L2-CAE	VAE	CAVGA	AnoViT	MKD	Ours
Carpet	67.0	49.0	54.0	67.0	73.0	50.0	79.3	**85.0**
Grid	69.0	51.0	78.0	83.0	75.0	52.0	78.1	**89.6**
Leather	46.0	52.0	77.0	71.0	71.0	85.0	**95.1**	92.0
Tile	52.0	51.0	85.0	81.0	70.0	89.0	91.6	**92.8**
Wood	83.0	68.0	**98.0**	89.0	85.0	95.0	94.3	95.3
Bottle	88.0	69.0	77.0	86.0	89.0	83.0	**99.4**	94.0
Cable	61.0	53.0	66.0	56.0	63.0	74.0	89.2	**93.0**
Capsule	61.0	58.0	67.0	77.0	83.0	73.0	81.5	**83.7**
Huzelnut	54.0	50.0	88.0	74.0	84.0	88.0	98.4	**100.0**
Metal nut	54.0	50.0	42.0	78.0	67.0	86.0	73.6	**89.5**
Pill	60.0	68.0	68.0	80.0	**88.0**	72.0	82.7	86.3
Screw	51.0	35.0	**100.0**	71.0	77.0	100.0	83.3	**100.0**
Toothbrush	74.0	57.0	41.0	89.0	91.0	74.0	92.2	**93.2**
Transistor	52.0	67.0	**88.0**	70.0	73.0	83.0	85.6	86.8
Zipper	80.0	59.0	71.0	67.0	87.0	73.0	**93.2**	89.7
Average	63.0	55.0	73.0	77.0	78.0	78.0	87.8	**91.2**

**Table 4 sensors-24-02440-t004:** Anomaly localization results on the MVTec AD dataset (pixel level AUROC (%), PRO (%)).

Category	AE_SSIM	AnoGAN	VAE	MKD	SCADN	VT_ADL	Ours
Carpet	(87.0, 64.7)	(54.0, 20.4)	(73.5, 50.1)	(—, 87.9)	(75.0, 85.0)	(—, 77.3)	(**88.4**, **88.0**)
Grid	(94.0, 84.9)	(58.0, 22.6)	(96.1, 22.4)	(—, 95.2)	(**97.7**, **96.8**)	(—, 87.1)	(97.2, 96.3)
Leather	(78.0, 56.1)	(64.0, 37.8)	(92.5, 63.5)	(—, 94.5)	(**99.3**, **98.7**)	(—, 72.8)	(96.6, 95.0)
Tile	(59.0, 17.5)	(50.0, 17.7)	(65.4, 87.0)	(—, 94.6)	(**96.7**, **95.3**)	(—, 79.6)	(92.8, 93.4)
Wood	(73.0, 60.5)	(62.0, 38.6)	(83.8, 62.8)	(—, 91.1)	(87.0, 85.3)	(—, 78.1)	(**91.4**, **90.0**)
Bottle	(93.0, 83.4)	(86.0, 62.0)	(92.2, 89.7)	(—, 93.1)	(**96.8**, 92.9)	(—, **94.9**)	(95.1, 92.1)
Cable	(82.0, 47.8)	(78.0, 38.3)	(91.0, 65.4)	(—, 81.8)	(89.2, 89.9)	(—, 77.6)	(**92.6**, **91.4**)
Capsule	(94.0, 86.0)	(84.0, 30.6)	(91.7, 52.6)	(—, **96.8**)	(86.0, 91.4)	(—, 67.2)	(**93.1**, 92.2)
Huzelnut	(97.0, 91.6)	(87.0, 69.8)	(97.6, 87.8)	(—, 96.5)	(97.1, 93.6)	(—, 89.7)	(**98.2**, **99.0**)
Metal nut	(89.0, 60.3)	(76.0, 32.0)	(90.7, 57.6)	(—, 94.2)	(**97.0**, **94.6**)	(—, 72.6)	(91.0, 89.0)
Pill	(91.0, 83.0)	(87.0, 77.6)	(93.0, 76.9)	(—, 96.1)	(**94.4**, **96.0**)	(—, 70.5)	(92.6, 93.2)
Screw	(96.0, 88.7)	(80.0, 46.6)	(94.5, 55.9)	(—, 94.2)	(87.0, 90.1)	(—, 92.8)	(**97.7**, **100.0**)
Toothbrush	(92.0, 78.4)	(90.0, 74.9)	(**98.5**, 69.3)	(—, 93.3)	(93.8, **90.7**)	(—, 90.1)	(89.4, 90.5)
Transistor	(90.0,72.5)	(80.0, 54.9)	(**91.9**, 62.6)	(—, 66.6)	(78.0, 75.3)	(—, 79.6)	(85.0, **81.0**)
Zipper	(88.0, 66.5)	(78.0, 46.7)	(86.9, 54.9)	(—, **95.1**)	(89.2, 89.2)	(—, 80.8)	(**93.2**, 94.0)
Average	(87.0, 69.4)	(74.3, 44.3)	(89.3, 63.9)	(—, 91.4)	(91.0, 90.4)	(—, 80.7)	(**93.0**, **92.3**)

**Table 5 sensors-24-02440-t005:** Anomaly detection and localization results on the BTAD dataset (image-level AUROC (%), pixel-level AUROC (%), PRO (%)).

Category	01	02	03	Average
P-SVDD	(95.7, 91.6, —)	(70.3, **92.7**, —)	(82.1, 91.0, —)	(82.7, 91.7, —)
SSPCAB	(96.2, 92.4, 62.8)	(69.3, 65.6, 28.6)	(**99.4**, 92.4, 71.0)	(88.3, 83.5, 54.1)
DRAEM	(98.5, 91.5, 61.4)	(68.6, 73.4, 39.0)	(98.6, 92.7, 84.3)	(88.6, 85.6, 61.6)
VT-ADL	(97.6, 76.3, 92.0)	(71.0, 88.9, **89.0**)	(82.6, 80.3, 86.0)	(83.7, 81.8, 89.0)
Ours	(**98.7**, **93.2**, **93.0**)	(**85.0**, 89.4, **89.0**)	(93.0, **93.1**, **91.0**)	(**92.2**, **91.9**, **91.0**)

**Table 6 sensors-24-02440-t006:** Results of ablation experiment on different modules used on the MVTecAD.

	M	CA	I-AUROC (%)	P-AUROC (%)	PRO (%)
Base			82.2	83.8	79.6
Base + M	√		85.7	91.8	90.5
Base + CA		√	84.5	86.4	85.7
Base + M + CA (Ours)	√	√	**91.2**	**93.0**	**92.3**

## Data Availability

The datasets used in this paper are public datasets.

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
