# Peer review of "An Unsupervised Method for Industrial Image Anomaly Detection with Vision Transformer-Based Autoencoder"

_sensors, 2024, doi:10.3390/s24082440_

Round 1

Reviewer 1 Report

Comments and Suggestions for Authors

The authors should revise the introduction, which is a bit shallow. The paper contains two Figures 1. The former should be edited to modify the known symbols to include English-based characters. The latter contains some typographical mistakes "Memory modoul".

The manuscript requires deep proofread and equations must be double-checked. For instance, mi. should be. $M_i$.

The bibliographical references are outdated, please revise literature and include references from last five or four years to validate that your research is novel.The proposed method is explained with the theoretical foundations, it is better to describe the foundations previously to describe the proposed method in detail.

The discussion obtained from the experimental results must be compared with additional literature outstanding methods. In fact, the authors are invited to include a subsection exclusive to describe all "Discussion"from results.

Tha captions in Figures and Tables, are quite short, please include a more detailed description of this important material.

The contribution should be highlighted while presenting the numerical results.

Comments on the Quality of English Language

Deep proofread is required, many typographical issues can be improved using a professional text editor based on latex.

Author Response

We are deeply grateful for the insightful feedback and constructive criticism offered by the reviewers. Each point raised has been meticulously considered, leading to targeted revisions of the manuscript. We are hopeful that these adjustments have refined our work to meet your expectations. Enclosed is our detailed response to the feedback provided.

Reviewer 2 Report

Comments and Suggestions for Authors

The paper is devoted to the development of a teacherless learning method for searching anomalous images based on a transform encoder. The results have scientific novelty and will be of interest to computer vision specialists. However, there are several remarks to the paper:

1) It would be interesting to mention specialized attacks on neural networks in addition to anomalies in the review (10.1016/j.procs.2021.04.170, 10.3390/photonics10020129).

2) Figure 1a probably has the character sequence written in Chinese (before the Position encoding block). Please provide an English version.

3) After formula (3), italicize the notations n, h, C, M.

4) Section 3: Figure 1 should be called Figure 2. There is a typo Memory modoul -> Memory module.

5) Check the whole numbering of the figures (figure 1 was encountered twice).

6) Replace the phrase after formula (4) with "encoder encodes" so that there is no indulgence.

7) Explain formulas (5) and (6) in more detail.

8) The numbering of formula (10) has moved to another line. Correction is needed.

9) Page 9: it seems that the image labels are missing in the sentence: "for the blocks centered at coordinates (i, j) in the original image and reconstructed image".

10) The numbering of formula (13) has moved to another line. Correction is needed.

11) The numbering of formula (15) has moved to another line. Correction is needed.

12) The paper gives the hardware characteristics of the calculator, but does not specify the training time and inference of the models.

Author Response

(The authors gave the same response as above.)

Reviewer 3 Report

Comments and Suggestions for Authors In the paper, an unsupervised industrial image anomaly detection method with vision transformer-based autoencoder is proposed. The authors analyzed current research works in the field and on this basis formulated a research problem. The effectiveness of the proposed approach was verified experimentally using publicly available datasets: MVTec AD and BeanTech AD. The efficacy of the proposed method was also compared with the results obtained by other state-of-the-art approaches.   Please address the following issues: 1. Figure 1 shows a fragment of the text that has not been translated into English. 2. Why were only two metrics chosen to compare the effectiveness of algorithms: AUROC and PRO? Please analyze what other indicators could be used and justify why these two were selected. 3. Please analyze the weaknesses of the proposed solution. In what cases will it be less effective than other methods? 4. Can the proposed approach also be used for other problems, or only for the problem discussed in the article? What could possibly need to be changed to use it in other tasks? Comments on the Quality of English Language

Please improve the English language of the article, especially in terms of style and grammar.

Author Response

(The authors gave the same response as above.)

Round 2

Reviewer 1 Report

Comments and Suggestions for Authors

The authors made an extraordinary work in addressing all my remarks correctly.

Comments on the Quality of English Language

The paper require a last proofreading.